# Adapted Physical Education: The Perspective of Asian Parents

**DOI:** 10.3390/ijerph19010091

**Published:** 2021-12-22

**Authors:** Eun Hye Kwon, Martin Block, Sean Healy, Tae-eung Kim

**Affiliations:** 1Department of Counseling, Health & Kinesiology, Texas A&M University-San Antonio, San Antonio, TX 78224, USA; eunhye.kwon@tamusa.edu; 2Department of Kinesiology, University of Virginia, Charlottesville, VA 22903, USA; meb7u@virginia.edu; 3School of Nursing, Psychotherapy and Community Health, Dublin City College, D09 Y5N0 Dublin, Ireland; sean.d.healy@dcu.ie; 4Department of Preventive Medicine, School of Medicine, Kyung Hee University, Seoul 02447, Korea

**Keywords:** adapted physical education, parents’ expectation, Asian parents

## Abstract

The purpose of this study was to examine the expectations from Adapted Physical Education services from the perspective of Asian parents (*n* = 8) who have children with disabilities. Data collection involved semi-structured interviews, completed in the participants’ preferred language. The data were analyzed using Braun and Clarke’s recipe for thematic analysis. Four themes emerged: (a) “overcoming” the disability in APE, (b) different perspectives on the importance of APE between mothers and fathers, (c) parents’ concern over children being “disrespectful,” and (d) communication issues. Since the culture in the United States is ethnically and socially more diversified, the significance and relevance of the results for effective, culturally sensitive APE provision is discussed. An increased understanding and involvement of Asian parents in terms of their children’s APE program will result in more culturally sensitive, effective, and relevant APE experiences.

## 1. Introduction

Nearly 7.3 million students receive special education services in the United States [1]. Such services are mandated in the Individuals with Disabilities Education Improvement Act of 2004. All students are required to receive physical education (PE) [2]. For some students with disabilities, for whom safe or successful participation in regular PE is not possible, adapted physical education (APE) services must be provided [2]. To control the quality of the APE program, the program is recommended to be provided by a qualified APE teacher [3].

It has been suggested that understanding family expectations of APE is of great importance and can aid the APE teachers by providing insight into the development of appropriate educational programs for the student and provide critical information for individualized educational plan (IEP) meetings and transition planning [4,5]. In addition, Martin and Smith indicated that communication from the APE teacher to the parents is necessary as parents should understand the purpose of APE and the role of APE teachers [6]. Research by Castaneda and Sherill reinforces the importance of the parent–teacher relationship, showing that in situations where coaches of a basketball league for children with disabilities interacted clearly with the parents, the families felt secure, confident, and empowered [7]. Indeed IDEA reflects the importance of parental input as it mandates consideration to be given to parental perspectives [2]. This is particularly important for families of students with disabilities as it is known that family expectations for the child will vary depending on the type and/or severity of the disability [8].

When we examine the research that focuses on parental perspectives, it becomes apparent that there is a dearth of literature on the parental perspectives of minority cultural groups. Currently, most of the studies addressing this topic primarily focus on the perspective of Caucasian families of children with disabilities [4,9]. Hispanic parents have had their perspectives heard, which, to the best of the authors’ knowledge, is the only research involving parents from minority cultural groups [5]. Yet the perspectives of such groups are important as cultural factors may shape the perspectives, beliefs, and values they have. It cannot be presumed that the perspectives of parents on children receiving APE services are generalizable across cultural groups. Thus, research into their perspectives is imperative for the development of culturally sensitive APE programs.

Research involving parents has provided an interesting perspective of APE. Caucasian parents of children with disabilities have shown to highly value the positive impact that PE may have on their children [6]. As a result, Caucasian parents tend to have certain expectations for their children and from the professionals who provide APE services for their children. Specific expectations of Caucasian families of children with disabilities included safety and successful participation in inclusive settings [10,11]. Downing and Rebollo conducted research exploring parental perceptions regarding the essential elements for the successful integration of a child with a disability into PE programs [12]. Parents indicated that the most critical elements for successful inclusion are the (a) physical educator’s motivation, (b) support from teachers and administrators, and (c) the health status of their children. The study also indicated that Caucasian parents also believe that family interests and expectations need to be considered for the successful implementation of an inclusive environment. Parents reported a desire to establish a collaborative relationship with the school to assure that their interests and expectations were valued [12]. Similarly, Russell explored the expectations of 10 Caucasian parents regarding APE. The findings indicated that expectations regarding APE and the transition process are (a) for their children to make progress in school, (b) for the parents to obtain support, and (c) to acquire information from professionals regarding the current level of fundamental motor skills [13].

Columna et al. interviewed 11 Hispanic parents of children with disabilities to investigate their expectations regarding APE teachers [5]. Hispanic parents expected that an increase in the level of independence in their children was a result of their participation in such programs; however, Hispanic parents also conveyed a sense of dissatisfaction with the professionals providing services for their children, primarily because they believed that the professionals did not always use their input. In addition, Hispanic parents wanted more information regarding APE, more communication with service providers, and more information in their native language. The key findings in the study were that Hispanic parents expect their children with disabilities to master the fundamental skills and participate in physical activities. However, communication issues are a barrier for parents to collaborate with APE teachers.

Such perspectives provide an interesting insight into APE and may help APE teachers to create culturally sensitive programs for their students. The results, however, cannot be thought to be generalizable to other cultural groups. Notably absent from the perspectives heard is that of Asian parents. Yet, numerous reasons exist as to why the perspectives of such a group are a necessary addition to the literature. Firstly, the U.S. Asian population is growing quickly, with a 72% increase between 2000 and 2015 [14]. With the growing number of people from Asian backgrounds present in the United States, there has been an increase in the number of Asian students with disabilities entering public school classrooms [1]. Thus, providing culturally appropriate instruction and collaborating with families has become increasingly important. Therefore, the purpose of this study was to understand the perspectives that Asian parents have on the APE services that their children receive.

## 2. Materials and Methods

### 2.1. Study Population and Sampling

A purposeful sampling strategy was implemented to recruit the 8 Asian parents that participated in this study. Recruitment began with emails first being sent from a professor of APE to APE teachers who are alumni of his APE program. The emails contained information sheets detailing the planned study, and the APE teachers were requested to print out the information sheet and give them to Asian parents of children who receive their APE services. The information sheet provided a basic description of the study and explained the commitment required for participation. The information sheet invited the parents to email or call the researcher if they were interested in participating in the study. Eight parents in total responded. At this point, the researcher ensured the individual met the sample criteria by reminding the participants of the eligibility criteria for participation: the individual should (1) be from an Asian country, (2) be educated in Asia, the education level being from elementary school through high school, and (3) have a child with a disability who is currently receiving APE services in a U.S. school. All individuals who wished to participate in the study met all criteria. The researchers’ institutional review board approved the research. A total of 8 participants agreed to be involved in the study; among the participants, there were 3 couples (Table 1).

### 2.2. Interview Procedure

Questions for the interview protocol were selected based on an extensive literature review, including an analysis of literature on APE, parental perspectives on APE and physical activity, and Asian parental perspectives on general education. Additionally, literature that investigated expectations of parents from diverse background for their children’s education was reviewed to design quality interview questions. A list of potential questions was compiled. To ensure content validity, a panel of experts, including professors in the fields of APE, PE, and family studies, reviewed the questions. The panel consisted of one professor whose area of specialty is APE, three nationally certified APE teachers who have had students from diverse ethnic backgrounds for longer than 5 years, and two certified physical education teachers who have had experience of including students with disabilities in his/her classes. These experts reviewed the original interview questions, suggesting the addition of a number of questions, including questions relating to the expected outcomes of the parents relating to the social and affective domains and questions regarding the parents’ perspectives on communication with the APE teachers. An information sheet regarding this study was distributed to local APE teachers. The APE teachers handed this information sheet to Asian parents. Those interested were directed to contact the primary investigator. Eligible participants then provided informed consent to participate in this study by signing an approved IRB consent form. All the participants took part in one semi-structured interview lasting approximately 45 min. The interview was conducted in each participant’s preferred language; one researcher spoke both English and Korean. A Chinese translator was used on one occasion, to translate both questions and answers. The primary investigator (PI), who speaks both English and Korean, conducted all interviews. Four interviews were conducted in Korean, three in English, and one in Chinese. All of the interviews were conducted in a mutually agreed upon location, free from distractions. A voice recorder recorded the interviews. The interviews were transcribed verbatim prior to analysis.

### 2.3. Data Analyses

An inductive thematic analysis was conducted so that the significance of a shared phenomenon could be revealed, providing a comprehensive account of data [15]. To ensure data analysis was undertaken in a methodological sound manner, the researchers used Braun and Clarke’s “recipe” for data analysis [15]. The first step involved gaining familiarization with the data. During transcription, initial notes were also made as suggested by Braun and Clarke [15]. Phase two of data analysis involved the generation of initial codes [15]. Quotes of interest and significance were highlighted on the transcripts and notes written derived from these data extracts, ensuring the analysis is data driven. Phase three then involved searching for themes, from the collated codes [15]. Four overarching themes emerged. Phase four involved a review of these themes. Through discussion between multiple researchers, a review of the relationship between themes was decided upon. A detailed analysis was then carried out on each theme to identify the “story” it told and how it fitted into the “broader overall story” [15]. Themes were then clearly defined. The final phase involved the write up of the thematic analysis, depicting the “complicated story,” assuring the reader of its merit and validity [15].

### 2.4. Trustworthiness

To ensure trustworthiness of the data where disagreement in interpretations arose, researchers discussed the issue until consensus was reached. In addition to interviews, field notes were kept and used to facilitate the analysis and interpretation of the information. After data analysis, member checking was used to ensure accurate interpretation of the participants’ perspectives was achieved. This was done through sending the results to and discussion of the research with all eight participants; both were sent in the language used in the interview. All participants were satisfied that the interpretation was correct.

## 3. Results

The purpose of this study was to examine the perspectives that Asian parents have on the APE services that their children receive. Four themes emerged from the analysis: (a) “overcoming” the disability in APE, (b) different perspectives between mothers and fathers, (c) parents’ concern over children being “disrespectful,” and (d) communication issues. The themes encapsulated various subthemes.

### 3.1. Overcoming the Disability in APE

Many of the parents interviewed spoke of APE in some positive terms. They tended to compare it to other areas of their children’s school life and noted its uniqueness in enabling their children to succeed and excel at something. Many parents felt this success was not being achieved in other areas of school life. The subthemes of “PE as a place to succeed” and “being like others” form this theme.

#### 3.1.1. PE as a Place to Succeed

A number of instances revealed an opinion that, in PE, the child with a disability could triumph over the challenges that affects him/her in other educational settings. PE was seen as a unique environment, less constraining and disabling than other classes. M2, a mother of a child with autism, signified this: “My son has difficulty to concentrate on given tasks. He is always in the resource room with his TA. However, in PE, it is different; he can try to concentrate harder and longer than in academic classes. Because it is the only time he can learn something using the physical domain, he does much better”. F3, a father spoke of a similar benefit: “He can do good in PE, much better than in other classes… and he knows it too, which is great.” Such observations were made by all the fathers interviewed and the one mother quoted above. They all recognized that their children were reaching their potential in PE.

#### 3.1.2. Be Like the Others

Some parents also spoke of how their child may perform similar to their peers in PE. PE was different from other aspects of education; it offered opportunities for their children to be “normal.” In the words of F1, a father of a student who was blind, “I hope my son could have a chance to overcome his disability in PE. Even though my son has visual impairment, I would like to see him run like his friends without disabilities. In PE, he can challenge himself to run as though he could see”. F2 echoing this opinion: “While running, they don’t see the autism; he can keep up with the others”. For some children of the interviewees, PE was an opportunity to fit in with the other classmates. It is unsure as to whether this was due to a well-implemented inclusive PE class or whether the children being discussed had good physical skills.

### 3.2. Differing Views

A clear dichotomy was evident in the interviews between the perspectives of mothers and fathers. A consistent difference of opinion existed especially concerning the value of PE.

#### 3.2.1. Fathers’ Perspectives

The fathers, on one hand, expressed an enthusiasm for PE and commented on its importance for their children, referring to the physical and social benefits of PE. For example, F3, a father of a student with autism, stated that “APE is extremely important to my son. My son is not good at English, and PE is the only time he could get along with his classmates. He can learn English and math at home”. F2 echoed such a view: “I think APE is as important as academic classes because, based on the skills he learned in APE, he can participate in sports after he graduates school. Also, I am worried that my son may get obese”. Such opinions clearly suggest a respect for PE and an awareness of its value for their children.

#### 3.2.2. Mothers’ Perspectives

The mothers, on the other hand, showed to hold PE in less esteem and prioritized other aspects of school life over it. They repeatedly voiced the opinion that for them, it is more important that their children focus on the academic subjects; PE was seen as a hindrance to learning in these more “important” areas. M1, the mother of a boy with autism, voiced this concern: “I hope my son will focus on academic goals. He is in Special Olympics every weekend, so it is enough for him. At the school, he must focus on reading and math because it is hard for me to teach these with my English”. M5 reflected this opinion: “It is hard to say that APE is so important. I would prefer him to work out at home with me; he can learn at school”. APE was clearly not seen as a place for “learning”. Such quotes signify the lack of educational value placed on PE, an opinion strongly shared by four of the mothers.

### 3.3. Perceived Burden of the Child

#### 3.3.1. Disturbance to Teachers

A number of parents were of the opinion that their children may be a “nuisance” in PE. It was common for the parents to speak of their concern that their son or daughter may be a distraction or an annoyance for the teacher due to his/her off-task behaviors. M2 exemplified this as she spoke of her daughter with autism: “I do not want to see her disturbing the class by screaming and laying down on the floor. It is always painful to hear that she pinched and scratched the teacher. It is not an appropriate attitude to the teacher”. Others, especially the parents of the students with autism, shared such a concern.

#### 3.3.2. Distraction to Peers

M2 also commented on how her daughter’s behavior may have a negative effect on the other students in the class: “Peers will also be distracted by my daughter...” F2 spoke of similar concerns regarding his son who has autism: “I am OK to see that my son could not participate in the activities. I believe that my son could learn something from the environment itself by listening and touching. However, I do not want to see him screaming and banging his head. It will negatively affect his peers”. Such comments imply an awareness of their children’s influence on the other students in the class. A number of parents were willing to have their children absent from participating in some PE activities to avoid causing a perceived negative impact on the PE environment for the other students.

### 3.4. Communication Issues

The parents in this study often spoke of the need for, and importance of, communication with the APE teachers. They showed a clear interest in being involved in their children’s APE progress. However, the majority of the parents interviewed spoke of challenges in achieving satisfactory communication. In particular, with reference to getting information about community physical activity opportunities for their children, the parents felt that they were unable to get such information from their PE teacher; F2’s comments exemplify this: “I am interested in community sports for my son. It is hard for me to get information about facilities and clubs for children with disabilities. I believe that the APE teacher knows that well, but it is hard to communicate with him”. M5 also raised this specific issue: “My daughter is interested in participating in sports. But I don’t know what sports are appropriate to her in terms of her disability and gender as lifetime sports. I would like to talk with the APE teacher to discuss about this”. The parents were slow to comment as to why they were unable to get such information from the APE teacher. M4, however, noted that it was a lack of time with the APE teacher that was the major barrier for her: “The only time I can talk to the APE teacher is the IEP meeting. Nevertheless, after the meeting, it is hard to see him. The teacher is always busy, on his way to teach”. M1 also reiterated this lack of communication, saying that “the report card every few months is just not enough”.

## 4. Discussion

The four themes that emerged reflect the issues of concern voiced by the Asian parents. A numbers of areas discussed echo opinions heard in other research on parental perspectives. However, a number of unique disclosures were also made.

### 4.1. Communication

The vital role of communication between parent and teacher for effective inclusive education has long been recognized and emphasized in literature [7,12,16]. Research on parental perspectives on PE reveals that parents have differing views on the effectiveness of this teacher–parent communication. The participants in this research consistently spoke of a need for increased communication. This is unsurprising, as it is known that parents are reluctant to contact teachers whose cultural backgrounds are different from their own in terms of language, religion, values, and country of origin [17]. For all participants in this study, English was their second language and this showed to be a barrier for communication with the PE teacher for a number of participants. For others, this language barrier was enlarged by a lack of time with the teacher, as demonstrated by the parents who spoke of the teacher rushing to class after the IEP meeting and of the infrequent report cards not being enough. Such frustration due to a lack of communication is shared by Asian parents interviewed on their perspectives on education in U.S. schools as well as Hispanic parents who called for more comprehensive communication with the APE teacher [5,17]. This lack of communication is particularly limiting to the information parents may receive regarding physical activity opportunities outside the school, as noted by a number of the interviewees. This was an issue of concern for parents of other studies also. Being a source of information about community physical activity opportunities is a role of the PE teacher, and it is a role that parents are repeatedly reporting as important yet unsatisfactorily addressed [3,5,18]. Although not as common, some Caucasian parents also noted a lack of communication as an issue [9]. A notable contrast between the opinions on communication of this sample and that of the Caucasian parents was that for the Asian parents, communication was seen as a one-way process; benefits would derive from communication from the teacher to the parent. Interestingly, the Caucasian parents stated that communication from them to the teachers was also of immense value, noting that they were advocates for their children and a source of information for the teacher. This key aspect of parent–teacher communication went unmentioned by the Asian parents in this study; this is maybe understandable as it has been noted that Asian parents tend to perceive the giving of ideas or suggestions to teachers as being disrespectful in that it challenges the teacher’s wisdom and authority [19]. PE teachers must work to inform the parents of their essential role as a source of important information that is imperative for the effective teaching of their children. A sample of Korean mothers have previously stated that they preferred indirect contact with teachers, such as email and school letters, as they understand written English better than spoken English. Such an approach may be of use to improve communication between PE teachers and parents also. One possible solution, offered by a parent interviewed in An and Goodwin’s study, was the use of a booklet for the exchange of notes between teacher and parent [9]. This may be of use in creating a line of communication between PE teacher and some parents of Asian background also.

Asian parents are interested in their children’s education and are ready to support their education. However, as most studies show, Asian parents are reluctant to communicate with the teachers in their children’s school because of language barriers and this lack of communication can lead to misunderstanding and be detrimental to the relationship with their children’s teachers. Researchers have found that families from Asia have different experiences in America compared with their experience in their home countries and they are sometimes dissatisfied with the education provided to their children [20]. Asian parents who have not been educated in the U.S. felt that their inability to teach their children was an obstacle to their children’s success. These parents were desperate for their children and wanted to receive information about how to best support their children [21].

### 4.2. Role of the PE Teacher

An interesting opinion, shared by half of the parents interviewed, was that their children may be burdensome or an annoyance to the teacher and a distraction to other students. Such a revelation was quite distinct from that in research on the perspectives of other parents. The parents did not believe it was the role of the PE teacher to deal with behaviors such as self-injurious actions and making noise. This belief is in stark contrast with other research, in which parents called for increased training for teachers to deal with their children’s needs and criticized teachers for lack of effort to accommodate their children; for example, parents in the study by Perkins et al. believed some teachers were too fixed in their routine and made inadequate effort to make accommodations to include their children [18]. Findings by Russell show that parents developed expectations concerning their children’s special education services based on their personal beliefs about disability and education [13]. Because of the strong belief in Confucianism, many parents from Asia will commonly hold teachers in high esteem; such admiration for the teacher leads to delegation of the children’s education entirely to the teacher [17]. Such a stance may result in the parents being reluctant to question the efforts of a teacher or be optimistic.

### 4.3. Parental Perspectives: Mother vs. Father

Within the three couples interviewed, a clear distinction can be seen between the perspectives of the fathers and mothers interviewed in this study. The fathers repeatedly commented on the social benefits their children may receive in PE; such opinions were reflected in research on PE involving Hispanic parents [5]. Parents of children with visual impairments also noted that the social benefits of physical activity have been of great importance for their children [18]. Mothers of children with spina bifida also recognized the importance of socialization in PE [9]. The mothers in this study, however, commonly voiced an opinion that PE was not a priority for their children, and the social benefits of the class were not recognized or deemed important. The mothers were of the opinion that the time would be better spent focusing on more academic subjects. Such a dichotomy of opinion has not been previously recognized in research on parental perspectives. The low esteem bestowed on PE, shown by the mothers in this study, may have a negative effect on the children’s belief about PE also; Stuart et al. remarked as to how parents of children with visual impairments held limiting beliefs and expectations about physical activity that their children may internalize; a similar worry holds true for the children of the mothers interviewed in this study [10]. Perhaps the lack of information about APE services that some parents have noted in other studies is extending for these mothers to a lack of information on the benefits of physical activity also [5]. This may present a challenge to PE teachers, who may have to inform the mothers of the benefits of physical activity.

The primary researcher conducted all interviews. The author’s prior knowledge and experience could bias the data collection and interpretation [22]. To reduce the effects of this, bracketing was completed prior to the interview process. This involved creating a bracketing mind-map to document the researcher’s perceptions, attitudes, and views, with the intent of moving those to the periphery so that participants’ views could be central to the analysis [23]. For example, bracketing notes in a reflective diary documented the author to be a proponent of inclusive APE. Effort was made to minimize the effect of identified biases on the analysis through consultation with the co-author.

This is was the first study to be conducted that examined the perspectives on APE in U.S. schools among parents of Asian background. A number of factors make the sample diverse and unique, and generalization of the results is not intended. Eight parents from eastern Asian countries participated in this research study; eastern Asia was chosen due to the similarities of culture and religion among its countries. However, future studies should examine the perspectives of parents from more specific countries or regions. In this study, all participants were highly educated, with a minimum of master’s degree. Education is an indicator of socioeconomic status. Since this education background may impact parents’ perspectives on APE (i.e., more highly educated parents may place a higher value on education), it is recommended that future research includes a more diverse sample to get a more generalized understanding of Asian parents’ perspectives on APE. The participant’s children had a range of disabilities, including autism and visual impairments. Future studies should examine how parental perspectives differ depending on the type of disability their children have. The ratio of mothers to fathers in the sample was 5:3. The separation of the view of fathers and mothers in the results shows a dichotomy between males and females; this should be examined in future studies involving bigger samples.

English was the second language of all participants. Although an interpreter was offered to all prior to the interviews, only one parent opted for this. Four interviews were done in Korean and three in English. The use of a second language for these three interviews resulted in shorter interviews with less depth. Future studies should seek to include interviewers with an ability to speak the interviewee’s first language to avoid this from becoming a barrier to communication. Acquiring in-depth information regarding parents’ background and type of disability their child has needs to be conducted prior to the interview for better understanding of parents. Additionally, it is recommended to separate the different perspectives of the father and the mother. It would provide a dichotomy between males and females, or fathers and mothers. Finally, comparing perspectives of parents from different ethnic backgrounds, such as American, European, Asian, and Hispanic, on APE will provide valuable understanding of the impact of cultural background for future studies.

The results of this study indicated that an increased understanding and involvement of Asian parents in terms of the implementation of their children’s APE program will result in more culturally sensitive, effective, and relevant PE experiences, since parents from different cultural backgrounds are sharing similar perspectives on the education of their children. Emerging themes of this study suggest that APE teachers need to have a basic understanding of students and their parents. Based on four themes that emerged from the analysis, (a) “overcoming” the disability in APE, (b) different perspectives between mothers and fathers, (c) parents’ concern over children being “disrespectful,” and (d) communication issues, specific communication strategies are recommended for Asian parents with children with disabilities. First, it is critical to communicate with both parents, mother and father, at the same time to drive common educational objectives enhancing the knowledge of the subject areas. The teachers need to provide detailed information to parents on the purposes and benefits of the subject area to promote their understanding regarding each subject area. Specifically in APE, the APE or PE teachers should clearly explain the goal of APE and physical activity (especially for mother), creating avenues for two-way communication between parent and teacher. When communicating with Asian parents with children with disabilities and who need translators, it is recommended to hire a translator with the relevant experience for providing information to such parents.

## Figures and Tables

**Table 1 ijerph-19-00091-t001:** Overview of participants.

Participant Number	Participant	Country of Origin	Child’s Gender	Child’s Age	Child’s Disability
M1	Mother	Korea	M	11	VI
M2	Mother	Korea	F	12	VI
M3	Mother	Korea	M	10	ASD
M4	Mother	China	M	14	ASD
M5	Mother	Japan	F	8	ASD
F1	Father	Korea	M	11	VI
F2	Father	Korea	M	12	ASD
F3	Father	China	F	14	ASD

Notes. ASD, autism spectrum disorder (ASD); VI, visual impairment.

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
