# Peer review of "Adapted Physical Education: The Perspective of Asian Parents"

_ijerph, 2021, doi:10.3390/ijerph19010091_

Round 1
Reviewer 1 Report
This manuscript describes the perspective of Asian parents for adaptive physical education of children with disabilities. The study was conducted on eight parents and the authors describe the four themes that emerged out from the analysis. Based on the themes, recommendations from authors will be helpful in the implementation of the children’s adaptive physical education in a more culturally-sensitive and effective way. Following are the comments/concerns that need to be addressed by authors:
- The language of the manuscript is clear and understandable.
- The abstract does not represent an accurate summary of the research. Authors can improve line 19 by adding the significance and relevance of the results.
- In line, 112 authors mentioned that 3 couples were involved in the study. Are these three couples contribute six participants out of eight? In this small sample size of eight where the Female: Male ratio is 5:3 concluding about the mother’s and father’s perspective is significant?
- Could authors explain lines 358, 359 about the overall level of education of participants? Are authors suggesting in this study that a higher level of education does not impart in deciding parent's perspective for adaptive physical education?
- There are grammatical mistakes in lines 353, 379 that need to be corrected.
- Please conclude the last paragraph (line 371-384) with a few lines on the overall impact of this study.
In my opinion, the manuscript entitled “Adapted Physical Education; the Perspective of Asian Parents” is suitable for publication in ‘International Journal of Environmental Research and Public health’ after the authors have addressed the comments.
Author Response
We are very much in honor of your valuable and go-to comments that helped make our paper better.
We attached the text and the revisions that we have made in the best possible words according to your comments.
Editor’s comments |
Response |
The language of the manuscript is clear and understandable |
The language of the manuscript was cleared by an editor whose first language is English.
|
The abstract does not represent an accurate summary of the research. Authors can improve line 19 by adding the significance and relevance of the results |
Line 19-22 Since the culture in the United States is ethnically and socially more diversified, the significance and relevance of the results for effective, culturally sensitive APE provision is discussed. An increased understanding and involvement of Asian parents in their children’s APE program will result in more culturally-sensitive, effective, and relevant APE experiences.
|
1.In line, 112 authors mentioned that 3 couples were involved in the study. Are these three couples contribute six participants out of eight? 2.In this small sample size of eight where the Female: Male ratio is 5:3 concluding about the mother’s and father’s perspective is significant?
|
1. Yes 2. Even though this study has a small sample size, we believe discussing the emergent differences in perspectives between mothers and fathers is important for informing future studies that may be better powered to examine differences in perspectives. |
Could authors explain lines 358, 359 about the overall level of education of participants? Are authors suggesting in this study that a higher level of education does not impart in deciding parent's perspective for adaptive physical education?
|
Line 369-374 In this study, all participants were highly educated, with a minimum of master’s degree. Education is an indicator of socioeconomic status. Since this education background may impact parents’ perspective toward APE (i.e., more highly educated parents may place a higher value on education), it is recommended that future research includes a more diverse sample to get a more generalized understanding of Asian parents’ perspectives of APE. |
There are grammatical mistakes in lines 353, 379 that need to be corrected. -This is was the first study to be conducted involving the perspective of parents of Asian background on inclusive PE in U.S. school. -The teachers need to provide detailed information on the purposes and benefits of the subject area. |
Line 364 This is was the first study to be conducted involving the perspective of parents of Asian background on inclusive PE in U.S. schools. Line 404 The teachers need to provide detailed information to parents on the purposes and benefits of the subject area to promote their understanding. |

Reviewer 2 Report
Congratulations for your interesting work. Hera yo have some consideration to improve it:
It's desiderable to know sample universe from which the final sample is obtained.
The inclusion criteria are very flexible, so that parents of children of very different ages are included, which surely means that the results are not uniform and, therefore, will weaken the conclusions.
It would be useful to provide more details on the development of the expert panel (how the final questions were decided, by what kind of agreements, etc.).
One way to enrich the content of the manuscript is to include a comparative table that allows the reader to know the different perspectives of families of Caucasian, Hispanic and Asian origin, certainly, the purpose of this study is to understand the perspective that Asian parents have on the APE services that their child receives.
- Considering that studies on the perspectives of parents of cultural minorities are scarce in the literature and that the sample is only 8 participating parents. This manuscript makes a contribution to the subject, considering that the cultural field requires deepening the values and beliefs of each particular group.
- Based on the above, it would be convenient to include a sample of the semi-structured interview that was used.
- I think the section of Results is very valuable, mainly the so-called: “overcoming the disability in APE”.
- The recommendations for future research projects allow us to know the possibilities of expansion of the initial research, so it seems to me very well marked.
1 How, future studies should examine how parental perspectives differ depending on the type of disability their child has.
2.the separation of the view of fathers and mothers in the results shows a dichotomy between males and females; this should be examined in future studies involving bigger samples.
3.future studies should seek to include interviewers with an ability to speak the interviewee´s first language to avoid this from becoming a barrier to communication… It is recommended to hire a translator with experience providing background information of parents with disabilities.
Good luck and thank you for your proposal!
Author Response
We are very much in honor of your valuable and go-to comments that helped make our paper better.
We attached the text and the revisions that we have made in the best possible words according to your comments.
Editor’s comments |
Response |
It's desirable to know sample universe from which the final sample is obtained.
|
Line 131 - An information sheet regarding this study was distributed to local APE teachers. The APE teachers handed this information sheet to Asian parents. Interested were directed to contact the primary investigator. Eligible participants then provided informed consent to participate in this study by signing an approved IRB consent form. |
The inclusion criteria are very flexible, so that parents of children of very different ages are included, which surely means that the results are not uniform and, therefore, will weaken the conclusions.
|
As it was discussed the purpose of this study is more focused on investigating parents’ expectations toward Adapted Physical Education (APE). That means all the children of the participants are included in a general education setting taking APE services. All the interview questions are fully focused on APE solely.
|
It would be useful to provide more details on the development of the expert panel (how the final questions were decided, by what kind of agreements, etc.). - Interview questions - Agreements were made via IRB approval - Penals
|
Interview questions: Line 118-122 - Questions for the interview protocol were selected based on an extensive literature review, including an analysis of literature on APE, parental perspectives in APE and physical activity, and Asian parental perspectives of general education. The additional literature review was conducted to verify valid interview questions that would effectively investigate expectations of parents, from different cultural backgrounds, expectation toward their children’s education. Agreements were made via IRB approval Line 112 Panels: Line 124-128 - The panel consisted of one professor whose area of specialties in APE, three APE nationally certified teachers who have had students with diverse ethnic backgrounds, and two certified Physical Education teachers who have had experiences to include students with disabilities in his/her classes |
One way to enrich the content of the manuscript is to include a comparative table that allows the reader to know the different perspectives of families of Caucasian, Hispanic and Asian origin, certainly, the purpose of this study is to understand the perspective that Asian parents have on the APE services that their child receives.
|
Line 389-392 - Finally, comparing perspectives toward APE of different ethnic backgrounds, such as American, European, Asian, and Hispanic, will provide a valuable understanding of the impact of cultural background for the future studies.
|
1 How, future studies should examine how parental perspectives differ depending on the type of disability their child has. 2.the separation of the view of fathers and mothers in the results shows a dichotomy between males and females; this should be examined in future studies involving bigger samples. 3.future studies should seek to include interviewers with an ability to speak the interviewee´s first language to avoid this from becoming a barrier to communication… It is recommended to hire a translator with experience providing background information of parents with disabilities.
|
Line 393-389 - Future studies should seek to include interviewers with the ability to speak the interviewee's first language to avoid this from becoming a barrier to communication. Acquiring background information regarding parents’ background and type of disability their child has needs to be conducted prior to the interview for better understanding toward parents. Additionally, it is recommended to separate the views on how different perspectives between father and mother. It would provide a dichotomy between males and females, or fathers and mothers.
|

Reviewer 3 Report
I consider that the content of the article is very important to determine the best strategies for collaboration with families. I find the cultural analysus great !
Congratulations !
It would be valuable to include a comparative chart between the perspectives of caucasian, hispanic and asdia families !
Author Response
We are very much in honour of your valuable and go-to comments that helped make our paper better.
We attached the text and the revisions that we have made in the best possible words according to your comments.
Editor’s comments |
Response |
It would be valuable to include a comparative chart between the perspectives of caucasian, hispanic and asdia families!
|
Line 389-392 - Finally, comparing perspectives toward APE of different ethnic backgrounds, such as American, European, and Hispanic, will provide valuable understanding the impact of cultural background for the future studies.
|
